# Sharpness Minimization Algorithms Do Not Only Minimize Sharpness To Achieve Better Generalization

**Kaiyue Wen**
Tsinghua University
wenky20@mails.tsinghua.edu.cn

Zhiyuan Li
Stanford University
zhiyuanli@stanford.edu

Tengyu Ma
Stanford University
tengyuma@stanford.edu

## Abstract

Despite extensive studies, the underlying reason as to why overparameterized neural networks can generalize remains elusive. Existing theory shows that common stochastic optimizers prefer flatter minimizers of the training loss, and thus a natural potential explanation is that flatness implies generalization. This work critically examines this explanation. Through theoretical and empirical investigation, we identify the following three scenarios for two-layer ReLU networks: (1) flatness provably implies generalization; (2) there exist non-generalizing flattest models and sharpness minimization algorithms fail to generalize poorly, and (3) perhaps most strikingly, there exist non-generalizing flattest models, but sharpness minimization algorithms still generalize. Our results suggest that the relationship between sharpness and generalization subtly depends on the data distributions and the model architectures and sharpness minimization algorithms do not only minimize sharpness to achieve better generalization. This calls for the search for other explanations for the generalization of over-parameterized neural networks.

## 1 Introduction

It remains mysterious why stochastic optimization methods such as stochastic gradient descent (SGD) can find generalizable models even when the architectures are overparameterized (Zhang et al., 2016; Gunasekar et al., 2017; Li et al., 2017; Soudry et al., 2018; Woodworth et al., 2020). Many empirical and theoretical studies suggest that generalization is correlated with or guaranteed by the flatness of the loss landscape at the learned model (Hochreiter & Schmidhuber, 1997; Keskar et al., 2016; Dziugaite & Roy, 2017; Jastrzebski et al., 2017; Neyshabur et al., 2017; Wu et al., 2018; Jiang et al., 2019; Blanc et al., 2019; Wei & Ma, 2019a,b; HaoChen et al., 2020; Foret et al., 2021; Damian et al., 2021; Li et al., 2021; Ma & Ying, 2021; Ding et al., 2022; Nacson et al., 2022; Wei et al., 2022; Lyu et al., 2022; Norton & Royset, 2021; Wu & Su, 2023). Thus, a natural theoretical question is

**Question 0.** *Does the flatness of the minimizers always correlate with the generalization capability?*

The answer to the question turns out to be false. First, Dinh et al. (2017) theoretically construct *very sharp* networks with good generalization. Second, recent empirical results (Andriushchenko et al., 2023b) find that sharpness may not have a strong correlation with test accuracy for a collection of modern architectures and settings, partly due to the same reason—there exist sharp models with good generalization. We note that, technically speaking, Question 0 is ill-defined without specifying the collection of models on which the correlation is evaluated. However, those sharp but generalizable models appear to be the main cause for the non-correlation.

37th Conference on Neural Information Processing Systems (NeurIPS 2023).

| Architecture | All Flattest Minimizers Generalize Well. | Sharpness Minimization Algorithms Generalize. |
|---|---|---|
| 2-layer w/o Bias | ✓ (Theorem 3.1) | ✓ |
| 2-layer w/ Bias | ✗ (Theorem 4.1) | ✗ |
| 2-layer w/ simplified BatchNorm | ✓ (Theorem 3.2) | ✓ |
| 2-layer w/ simplified LayerNorm | ✗ (Theorem 5.1) | ✓ |

Table 1: **Overview of Our Results.** Each row in the table corresponds to one architecture. The second column indicates whether all flattest minimizers of training loss generalize well. ✓ indicates that all (near) flattest minimizers of training loss provably generalize well and ✗ indicates that there provably exists flattest minimizers that generalize poorly. The third column indicates whether the sharpness minimization algorithms generalize well in our experiments. Results in row 2 and 4 deny Question 1 and Question 2 respectively.

Observing the existing theoretical and empirical evidence, it is natural to ask the one-side version of Question 0, where we are only interested in whether sharpness implies generalization but not vice versa.

**Question 1.** *Do all the flattest neural network minimizers generalize well?*

Though there are some theoretical works that answer Question 1 affirmatively for simplified linear models (Li et al., 2021; Ding et al., 2022; Nacson et al., 2022; Gatmiry et al., 2023), the answer to Question 1 for standard neural networks remains unclear. Those theoretical results linking generalization to sharpness for more general architectures typically also involve other terms in generalization bounds, such as parameter dimension or norm (Neyshabur et al., 2017; Foret et al., 2021; Wei & Ma, 2019a,b; Norton & Royset, 2021), thus do not answer Question 1 directly.

Our first contribution is a theoretical analysis showing that the answer to Question 1 can be **false**, even for simple architectures like 2-layer ReLU networks. Intriguingly, we also find that the answer to Question 1 subtly depends on the architectures of neural networks. For example, simply removing the bias in the first layer turns the aforementioned negative result into a positive result, as also shown in the Theorem 4.3 of Wu & Su (2023) (that the authors only came to be aware of after putting this work online).

More concretely, we show that for the 2 parity xor problem with mean square loss and with data sampled from hypercube $\{-1, 1\}^d$, all flattest 2-layer ReLU neural networks without bias provably generalize. However, when bias is added, for the same data distribution and loss function, there exists a flattest minimizer that fails to generalize for every unseen data. Since adding bias in the first layer can be interpreted as appending a constant input feature, this result suggests that the generalization of the flattest minimizer is sensitive to both network architectures and data distributions.

Recent theoretical studies (Wu et al., 2018; Blanc et al., 2019; Damian et al., 2021; Li et al., 2021; Arora et al., 2022; Wen et al., 2022; Nacson et al., 2022; Lyu et al., 2022; Bartlett et al., 2022; Li et al., 2022) also show that optimizers including SGD with large learning rates or label noise and Sharpness-Aware Minimization (SAM, Foret et al. (2021)) may implicitly regularize the sharpness of the training loss landscape. These optimizers are referred to as *sharpness minimization algorithms* in this paper. Because Question 1 is not always true, it is then natural to hypothesize that sharpness-minimization algorithms will fail for architectures and data distributions where Question 1 is not true.

**Question 2.** *Will sharpness minimization algorithm fail to generalize when there exist non-generalizing flattest minimizers?*

A priori, the authors were expecting that the answer to Question 2 is affirmative, which means that a possible explanation is that the sharpness minimization algorithm works if and only if for certain architecture and data distribution, Question 1 is true. However, surprisingly, we also answer this question negatively for some architectures and data distributions. In other words, we found that sharpness-minimization algorithms can still generalize well even when the answer to Question 1 is false. The result is consistent with our theoretical discovery that for many architectures, there exist both non-generalizing and generalizing flattest minimizers of the training loss. We show empirically that sharpness-minimization algorithms can find different types of minimizers for different architectures.

Our results are summarized in Table 1. We show through theoretical and empirical analysis that the relationship between sharpness and generalization can fall into three different regimes depending on the architectures and distributions. The three regimes include:

- **Scenario 1.** Flattest minimizers of training loss provably generalize and sharpness minimization algorithms find generalizable models. This regime (Theorems 3.1 and 3.2) includes 2-layer ReLU MLP without bias and 2-layer ReLU MLP with a simplified BatchNorm (without mean subtraction and bias). We answer both the Question 1 and Question 2 affirmatively in this scenario.[1]
- **Scenario 2.** There exists a flattest minimizer that has the worst generalization over all minimizers. Also, sharpness minimization algorithms fail to find generalizable models. This regime includes 2 layer ReLU MLP with bias. We deny Question 1 while affirm Question 2 in this scenario.
- **Scenario 3.** There exist flattest minimizers that do not generalize but the sharpness minimization algorithm still finds the generalizable flattest model empirically. This regime includes 2-layer ReLU MLP with a simplified LayerNorm (without mean subtraction and bias). In this scenario, the sharpness minimization algorithm relies other unknown mechanisms beyond minimizing sharpness to find a generalizable model. We deny both Question 1 and Question 2 in this scenario.

## 2 Setup

**Rademacher Complexity.** Given $n$ data $S = \{x_i\}_{i=1}^n$, the *empirical Rademacher complexity* of function class $\mathcal{F}$ is defined as $\mathcal{R}_S(\mathcal{F}) = \frac{1}{n} \mathbb{E}_{\epsilon \sim \{\pm 1\}^n} \sup_{f \in \mathcal{F}} \sum_{i=1}^n \epsilon_i f(x_i)$. **Architectures.** As summarized in Table 1, we will consider multiple network architectures and discuss how architecture influences the relationship between sharpness and generalization. For each model $f_\theta$ parameterized by $\theta$, we will use $d$ to denote the input dimension and $m$ to denote the network width. We will now describe the architectures in detail.

**2-MLP-No-Bias.** $f_\theta^{\mathrm{nobias}}(x) = W_2\mathsf{relu}\,(W_1 x)$ with $\theta = (W_1, W_2)$.

**2-MLP-Bias.** $f_\theta^{\mathrm{bias}}(x) = W_2\mathsf{relu}\,(W_1 x + b_1)$ with $\theta = (W_1, b_1, W_2)$. We additionally define MLP-Bias as $f_\theta^{\mathrm{bias,D}}(x) = W_D\mathsf{relu}\cdots W_2\mathsf{relu}\,(W_1 x + b_1)$,

**2-MLP-Sim-BN.** $f_\theta^{\mathrm{sbn}}(x, \{x_i\}_{i \in [n]}) = W_2\mathsf{SBN}_\gamma\,(\mathsf{relu}\,(W_1 x + b_1), \{\mathsf{relu}\,(W_1 x_i + b_1)\})$, where the simplified BatchNorm SBN is defined as $\forall m, n \in N, \forall i \in [n], x, x_i \in \mathbb{R}^m, j \in [m], \mathsf{SBN}_\gamma(x, \{x_i\}_{i \in [n]})[j] = \gamma x[j]/ \left(\sum_{i=1}^n (x_i[j])^2/n\right)^{1/2}$ and $\theta = (W_1, b_1, \gamma, W_2)$.

**2-MLP-Sim-LN.** $f_\theta^{\mathrm{sln}}(x) = W_2 \frac{\mathsf{relu}(W_1 x + b_1)}{\max\{\|\mathsf{relu}(W_1 x + b_1)\|_2, \epsilon\}}$ where $\epsilon$ is a sufficiently small positive constant.

Surprisingly, our results show that the relationships between sharpness and generalization are strikingly different among these simple yet similar architectures.

**Data Distribution.** We will consider a simple data distribution as our testbed. Data distribution $\mathcal{P}_{\mathrm{xor}}$ is a joint distribution over data point $x$ and label $y$. The data point is sampled uniformly from the hypercube $\{-1, 1\}^d$ and the label satisfies $y = x[1]x[2]$. Many of our results, including our generalization bound in Section 3 and experimental observations can be generalized to broader family of distributions (Appendix B).

**Loss.** We will use mean squared error $\ell_{\mathrm{mse}}$ for training and denote the training loss as $L$. In Appendix B, we will show that all our theoretical results and empirical observations hold for logistic loss with label smoothing probability $p > 0$. We will also consider zero one loss $\Pr(y f_\theta(x) > 0)$ for evaluating the model. We will use interpolating model to denote the model with parameter $\theta$ that minimizes $L$.

**Definition 2.1** (Interpolating Model). *A model $f_\theta$ interpolates the dataset $\{(x_i, y_i)\}_{i=1}^n$ if and only if $\forall i, f_\theta(x_i) = y_i$.*

**Sharpness.** Our theoretical analysis focuses on understanding the *sharpness* of the trained models. Precisely, for a model $f_\theta$ parameterized by $\theta$, a dataset $\{(x_i, y_i)\}_{i=1}^n$ and loss function $\ell$, we will use the trace of Hessian of loss function, $\mathrm{Tr}(\nabla^2 L(\theta))$ to measure how sharp the loss is at $\theta$, which is a proxy for the sharpness along a random direction (Wen et al., 2022), or equivalently, the expected increment of loss under a random gaussian perturbation (Foret et al., 2021; Orvieto et al., 2022).

$\mathrm{Tr}(\nabla^2 L(\theta))$ is not the only choice for defining sharpness, but theoretically many sharpness minimization algorithms have been shown to minimize this term over interpolating models. In particular,

---

[1]The condition for Question 2 is not satisfied and thus the answer to Question 2 is affirmative.

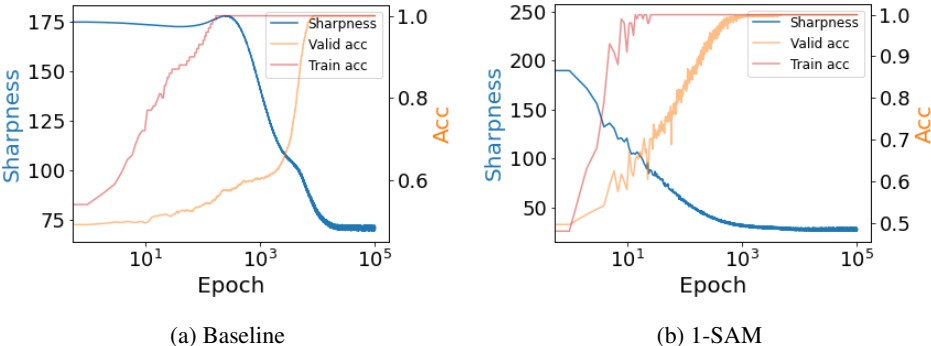

(a) Baseline        (b) 1-SAM

Figure 1: **Scenario I.** We train a 2-layer MLP with ReLU activation without bias using gradient descent with weight decay and 1-SAM on $\mathcal{P}_{\text{xor}}$ with dimension $d = 30$ and training set size $n = 100$. In both cases, the model reaches perfect generalization. Notice that although weight decay doesn't explicitly regularize model sharpness, the flatness of the model decreases through training, which is consistent with our Lemma 3.1 relating sharpness to the norm of the weight.

under the assumptions that the minimizer of the training loss form a smooth manifold Cooper (2018); Fehrman et al. (2020), Sharpness-Aware Minimization (SAM) (Foret et al., 2021) with batch size 1 and sufficiently small learning rate $\eta$ and perturbation radius $\rho$ (Wen et al., 2022; Bartlett et al., 2022), or Label Noise SGD with sufficiently small learning rate $\eta$ (Blanc et al., 2019; Damian et al., 2021; Li et al., 2021), prefers interpolating models with small trace of Hessian of the loss. Hence, we choose to analyze trace of Hessian of the loss and will use SAM with batch size 1 (we denote it by 1-SAM) as our sharpness minimization algorithm in our experiments.

**Notations.** We use Tr to denote the trace of a matrix and $x[i]$ to denote the value of the $i$-th coordinate of vector $x$. We will use $\odot$ to represent element-wise product. We use $\mathbf{1}$ as the (coordinate-wise) indicator function, for example, $\mathbf{1}\left[x > 0\right]$ is a vector of the same length as $x$ whose $j$-th entry is 1 if $x[j] > 0$ and 0 otherwise. We will use $\tilde{O}(x)$ to hide logarithmic multiplicative factors.

## 3 Scenario I: All Flattest Models Generalize

### 3.1 Flattest models provably generalize

When the architecture is **2-MLP-No-Bias**, we will show that the flattest models can provably generalize, hence answering Question 1 affirmatively for this architecture and data distribution $\mathcal{P}_{\text{xor}}$.

**Theorem 3.1.** *For any* $\delta \in (0, 1)$ *and input dimension* $d$*, for* $n = \Omega\left(d\log\left(\frac{d}{\delta}\right)\right)$*, with probability at least* $1 - \delta$ *over the random draw of training set* $\{(x_i, y_i)\}_{i=1}^n$ *from* $\mathcal{P}_{\text{xor}}^n$*, let* $L(\theta) \triangleq \frac{1}{n}\sum_{i=1}^n \ell_{\text{mse}}(f_\theta^{\text{nobias}}(x_i), y_i)$ *be the training loss for* **2-MLP-No-Bias***, it holds that for all* $\theta^* \in \arg\min_{L(\theta)=0} \text{Tr}\left(\nabla^2 L\left(\theta\right)\right)$*, we have that*

$$\mathrm{E}_{x,y\sim\mathcal{P}_{\text{xor}}}\left[\ell_{\text{mse}}\left(f_{\theta^*}^{\text{nobias}}\left(x\right), y\right)\right] \leq \tilde{O}\left(d/n\right).$$

Theorem 3.1 shows that for $\mathcal{P}_{\text{xor}}$, flat models can generalize under almost linear sample complexity with respect to the input dimension. We note that Theorem 3.1 implies that $\Pr_{x,y\sim\mathcal{P}_{\text{xor}}}\left[f_{\theta^*}^{\text{nobias}}(x)y > 0\right] \leq \tilde{O}\left(d/n\right)$. because if $f_{\theta^*}^{\text{nobias}}(x)y \leq 0$, it holds that $\ell_{\text{mse}}\left(f_{\theta^*}^{\text{nobias}}\left(x\right), y\right) \geq 1$. This shows that the model can classify the input with high accuracy. The major proof step is relating sharpness to the norm of the weight itself.

**Lemma 3.1.** *Define* $\Theta_C \triangleq \{\theta = (W_1, W_2) \mid \sum_{j=1}^m \|W_{1,j}\|_2|W_{2,j}| \leq C\}$*. Under the setting of Theorem 3.1, there exists a absolute constant* $C$ *independent of* $d$ *and* $\delta$*, such that with probability at least* $1 - \delta$*,* $\arg\min_{L(\theta)=0} \text{Tr}\left(\nabla^2 L\left(\theta\right)\right) \subseteq \Theta_C$ *and* $\mathcal{R}_S(\{f_\theta^{\text{nobias}} \mid \theta \in \Theta_C\}) \leq \tilde{O}\left(\sqrt{d/n}\right)$*.*

We would like to note that similar results of Theorem 3.1 and lemma 3.1 have also been shown in a prior work Wu & Su (2023) (that the authors were not aware of before the first version of this work was online).

The almost linear complexity in Theorem 3.1 is not trivial. For example, Wei et al. (2019) shows that learning the distribution will require $\Omega(d^2)$ samples for Neural Tangent Kernel (NTK) (Jacot et al.,

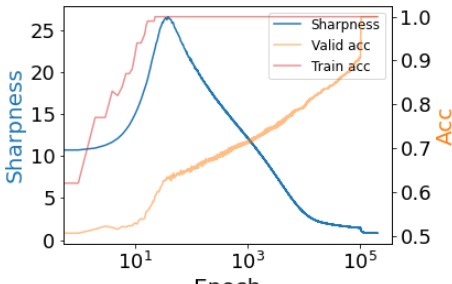

| $W_{1,i}[1]$ | $W_{1,i}[2]$ | $\|W_{1,i}[3:d]\|_2$ |
|---|---|---|
| 18.581 | -18.582 | 0.02 |
| -14.363 | -14.363 | 0.03 |
| 13.768 | 13.771 | 0.03 |
| -12.601 | 12.601 | 0.01 |

(a) 2-layer MLP with simplified BN

(b) Weights of the four neurons with the largest norm in the first Layer

Figure 2: **Interpretable Flattest Solution** We train a 2-layer MLP with simplified BN using 1-SAM on $\mathcal{P}_{\mathrm{xor}}$ with dimension $d = 30$ and training set size $n = 100$. After training, we find that the model is indeed interpretable. In Figure 2b, we inspect the weight of the four neurons of the four largest neurons in the first layer and we observe that the four neurons approximately extract features $\pm x[1] \pm x[2]$.

2018). In contrast, our result shows that learning the distribution only requires $\tilde{O}(d)$ samples as long as the flatness of the model is controlled.

Beyond reducing model complexity, flatness may also encourage the model to find a more interpretable solution. We prove that under a stronger than i.i.d condition over the training set, the near flattest interpolating model with architecture 2-MLP-Sim-BN will provably generalize and the weight of the first layer will be centered on the first two coordinates of the input, i.e., $\|W_{1,i}[3:d]\|_2 \leq \epsilon\|W_{1,i}\|_2$.

**Condition 1** (Complete Training Set Condition). *There exists set $S \subset \{-1, 1\}^{d-2}$, such that the linear space spanned by $S - S = \{s_1 - s_2 \mid s_1, s_2 \in S\}$ has rank $d - 2$ and the training set is $\{(x, y) \mid x \in \mathbb{R}^d, x[3:d] \in S, x[1], x[2] \in \{-1, 1\}, y = x[1] \times x[2]\}$.*

**Theorem 3.2.** *Given any training set $\{(x_i, y_i)\}_{i=1}^n$ satisfying Condition 1, for any width $m$ and any $\epsilon > 0$, there exists constant $\kappa > 0$, such that for any width-$m$ 2-MLP-Sim-BN, $f^{\mathrm{sbn}}$, satisfying $f_\theta^{\mathrm{sbn}}$ interpolates the training set and $\mathrm{Tr}\left(\nabla^2 L(\theta)\right) \leq \kappa + \inf_{L(\theta')=0} \mathrm{Tr}\left(\nabla^2 L(\theta')\right)$, it holds that $\forall x \in \{-1, 1\}^d, \left|x[1]x[2] - f_\theta(x)\right| \leq \epsilon$ and that $\forall i \in [m], \|W_{1,i}[3:d]\|_2 \leq \epsilon\|W_{1,i}\|_2$.*

One may notice that in Theorem 3.2 we only consider the approximate minimizer of sharpness. This is because the gradient of output with respect to $W_1, b_1$, despite never being zero, will converge to zero as the norm of $W_1, b_1$ converges to $\infty$.

Condition 1 may seem stringent. In practice (Figure 2b), we find it not necessary for 1-SAM to find a generalizable solution. We hypothesize that this condition is mainly technical. Theorem 3.2 shows that sharpness minimization may guide the model to find an interpretable and low-rank representation. Similar implicit bias of SAM has also been discussed in Andriushchenko et al. (2023a) The proof is deferred to Appendix B.1

## 3.2 SAM empirically finds the flattest model that generalizes

We use 1-SAM to train 2-MLP-No-Bias on data distribution $\mathcal{P}_{\mathrm{xor}}$ to verify our Theorem 3.1 (Figure 1). As expected, the model interpolates the training set and reaches a flat minimum that generalizes perfectly to the test set.

We then verify our Theorem 3.2 by training a 2-layer MLP with simplified BN on data distribution $\mathcal{P}_{\mathrm{xor}}$ (Figure 2a). Here we do not enforce the strong theoretical Condition 1. However, we still observe that SAM finds a flat minimum that generalizes well. We then perform a detailed analysis of the model and find that the model is indeed interpretable. For example, the four largest neurons in the first layer approximately extract features $\{\mathrm{relu}(c_1 x[1] + c_2 x[2]) \mid c_1, c_2 \in \{-1, 1\}\}$ (Figure 2b). Also, the first 2 columns of the weight matrix of the first layer, corresponding to the useful features $\{\mathrm{relu}(c_1 x[1] + c_2 x[2]) \mid c_1, c_2 \in \{-1, 1\}\}$, have norms 42.47 and 42.48, while the largest column norm of the rest of the weight matrix is only 5.65.

# 4  Scenario II: Both Flattest Generalizing and Non-generalizing Models Exist, and SAM Finds the Former

## 4.1  Both generalizing and non-generalizing solutions can be flattest

In previous section, we show through Theorems 3.1 and 3.2 that sharpness benefits generalization under some assumptions. It is natural to ask whether it is possible to extend this bound to general architectures. However, in this section, we will show that the generalization benefit depends on model architectures. In fact, simply adding bias to the first layer of 2-MLP-No-Bias makes non-vacuous generalization bound impossible for distribution $\mathcal{P}_{\text{xor}}$. This then leads to a negative answer to Question 1.

**Definition 4.1** (Set of extreme points). *A finite set $S \subset \mathbb{R}^d$ is a set of extreme points if and only if for any $x \in S$, $x$ is a vertex of the convex hull of $S$.*

**Definition 4.2** (Memorizing Solutions). *A $D$-layer network is a* memorizing solution *for a training dataset if (1) the network interpolates the training dataset, and (2) for any depth $k \in [D-1]$, there is an injection from the input data to the neurons on depth $k$, such that the activations in layer $k$ for each input data is a one-hot vector with the non-zero entry being the corresponding neuron.*

**Theorem 4.1.** *For any $D \geq 2$, if the input data points $\{x_i\}$ of the training set form a set of extreme points (Definition 4.1), then there exists a width $n$ layer $D$ MLP-Bias that is a memorizing solution (Definition 4.2) for the training dataset and has minimal sharpness over all the interpolating solutions.*

As one may suspect, these memorizing solutions can have poor generalization performance.

**Proposition 4.1.** *For data distribution $\mathcal{P}_{\text{xor}}$, for any number of samples $n$, there exists a width-$n$ 2-MLP-Bias that memorizes the training set as in Theorem 4.1, reaches minimal sharpness over all the interpolating models and has generalization error $\max\{1 - n/2^d, 0\}$ measured by zero one error.*

This corollary shows that a flat model can generalize poorly. Comparing Theorems 3.1 and 4.1, one may observe the perhaps surprising difference caused by slightly modifying the architectures (adding bias or removing the BatchNorm). To further show the complex relationship between sharpness and generalization, the following theorem suggests, despite the existence of memorizing solutions, there also exists a flattest model that *can* generalize well.

**Proposition 4.2.** *For data distribution $\mathcal{P}_{\text{xor}}$, for any number of samples $n$, there exists a width-$n$ 2-MLP-Bias that interpolates the training dataset, reaches minimal sharpness over all the interpolating models, and has zero generalization error measured by zero one error.*

The flat solution constructed is highly simple. It contains four activated neurons, each corresponding to one feature in $\pm x[1] \pm x[2]$ (Equation (5)).

**Proof sketch.** For simplicity, we will consider 2-MLP-Bias here. The construction of the memorizing solution in Theorem 4.1 is as follows (visualized in Figure 3). As the input data points form a set of extreme points (Definition 4.1), for each input data point $x_i$, there exists a vector $\|w_i\| = 1, w_i \in \mathbb{R}^d$, such that $\forall j \neq i, w_i^\top x_i > w_i^\top x_j$. We can then choose

$$W_1 = [\sqrt{r_i|y_i|}w_i/\epsilon]_i^\top, b_1 = [\sqrt{r_i|y_i|}\left(-w_i^\top x_i + \epsilon\right)/\epsilon]^\top, W_2 = [\text{sign}(y_i)\sqrt{|y_i|/r_i}]_i.$$

Here $r_i = (\|x_i\|^2 + 1)^{1/2}$ and $\epsilon$ is a sufficiently small positive number. Then it holds that $\text{relu}(W_1 x_i + b_1) = \sqrt{r_i|y_i|}e_i$, where $e_i$ is the $i-$th coordinate vector. This shows there is a one-to-one correspondence between the input data and the neurons. It is easy to verify that the model interpolates the training dataset. Furthermore, for $\mathcal{P}_{\text{xor}}$ and sufficiently small $\epsilon$, for any input $x \notin \{x_i\}_{i \in [n]}$, it holds that $\text{relu}(W_1 x + b_1) = 0$. Hence the model will output the same label 0 for all the data points outside the training set. This indicates Proposition 4.1.

To show the memorization solution has minimal sharpness, we need the following lemma that relates the sharpness and the Jacobian of the model.

**Lemma 4.1.** *For mean squared error loss $l_{mse}$, if model $f_\theta$ is differentiable and interpolates dataset $\{(x_i, y_i)\}_{i \in [n]}$, then $\text{Tr}\left(\nabla^2 L(\theta)\right) = \frac{2}{n}\sum_{i=1}^{n}\|\nabla_\theta f_\theta(x_i)\|^2$.*

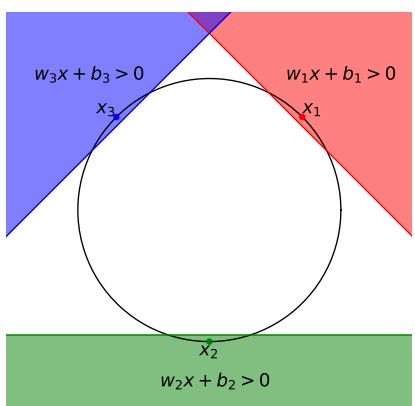

Figure 3: **Visualization of Memorization Solutions.** This is an illustration of the memorizing solutions constructed in Theorem 4.1. Here the input data points come from a unit circle and are marked as dots. The shady area with the corresponding color represents the region where the corresponding neuron is activated. One can see that the network can output the correct label for each input data point in the training set as long as the weight vector on the corresponding neuron is properly chosen. Further, the network will make the same prediction $0$ for all the input data points outside the shady area and this volume can be made almost as large as the support of the training set by choosing $\epsilon$ sufficiently small. Hence the model can interpolate the training set while generalizing poorly.

*Proof of Lemma 4.1.* By standard calculus, it holds that,

$$
\mathrm{Tr}\left(\nabla^2 L(\theta)\right) = \frac{1}{n}\sum_{i=1}^{n}\mathrm{Tr}\left(\nabla_\theta^2\left[(f_\theta(x_i) - y_i)^2\right]\right)
$$

$$
= \frac{2}{n}\sum_{i=1}^{n}\mathrm{Tr}\left(\nabla_\theta^2 f_\theta(x_i)(f_\theta(x_i) - y_i) + (\nabla_\theta f_\theta(x_i))(\nabla_\theta f_\theta(x_i))^\top\right)
$$

$$
= \frac{2}{n}\sum_{i=1}^{n}\mathrm{Tr}\left((\nabla_\theta f_\theta(x_i))(\nabla_\theta f_\theta(x_i))^\top\right) = \frac{2}{n}\sum_{i=1}^{n}\|\nabla_\theta f_\theta(x_i)\|_2^2. \tag{1}
$$

The first equation in Equation (1) use $\forall i, f_\theta(x_i) = y_i$. The proof is then complete. $\square$

After establishing Lemma 4.1, one can then explicitly calculate the lower bound of $\|\nabla_\theta f_\theta(x_i)\|^2$ condition on $f_\theta(x_i) = y_i$. For simplicity of writing, we will view the bias term as a part of the weight matrix by appending a 1 to the input data point. Precisely, we will use notation $x_i' \in \mathbb{R}^{d+1}$ to denote transformed input satisfying $\forall j \in [d], x_i'[j] = x_i[j], x_i'[d+1] = 1$ and $W_1' = [W_1, b_1] \in \mathbb{R}^{m\times(d+1)}$ to denote the transformed weight matrix.

By the chain rule, we have,

$$
\|\nabla_\theta f_\theta(x_i)\|^2 = \|\nabla_{W_1'} f_\theta(x_i)\|_F^2 + \|\nabla_{W_2} f_\theta(x_i)\|_F^2
$$

$$
= \|(W_2 \odot \mathbf{1}\left[W_1' x_i' > 0\right])x_i'^\top\|_F^2 + \|\mathsf{relu}\left(W_1' x_i'\right)\|_2^2.
$$

$$
= \|W_2 \odot \mathbf{1}\left[W_1' x_i' > 0\right]\|_2^2\|x_i'\|^2 + \|\mathsf{relu}\left(W_1' x_i'\right)\|_2^2. \tag{2}
$$

Then by Cauchy-Schwarz inequality, we have

$$
\|\nabla_\theta f_\theta(x_i)\|^2 = \|W_2 \odot \mathbf{1}\left[W_1' x_i' > 0\right]\|_2^2\|x_i'\|^2 + \|\mathsf{relu}\left(W_1' x_i'\right)\|_2^2
$$

$$
\geq 2\|x_i'\|\left|(W_2 \odot \mathbf{1}\left[W_1 x_i > 0\right])^\top \mathsf{relu}\left(W_1' x_i'\right)\right| = 2\|x_i'\|\,|y_i|. \tag{3}
$$

In Equation (3), we use condition $f_\theta(x_i) = y_i$. Finally, notice that the lower bound is reached when

$$
W_2 \odot \mathbf{1}\left[W_1' x_i' > 0\right] = \mathsf{relu}\left(W_1' x_i'\right)/\|x_i'\|. \tag{4}
$$

Condition Equation (4) is clearly reached for the memorization construction we constructed, where both sides of the equation are equal to $\sqrt{|y_i|/\|x_i'\|}e_i$. This completes the proof of Theorem 4.1.

However, the memorization network is not the only parameter that can reach the lower bound. For example, for distribution $\mathcal{P}_{\mathrm{xor}}$, if parameter $\theta$ satisfies,

$$
\forall i, j \in \{0, 1\}, W_{1,2i+j+1} = r[(-1)^i, (-1)^j, ..., 0], b_1[2i+j+1] = -r, W_2[2i+j] = (-1)^{i+j}/r. \tag{5}
$$

$$
\forall k > 4, W_{1,k} = [0, ..., 0], b_1[k] = 0, W_2[k] = 0,
$$

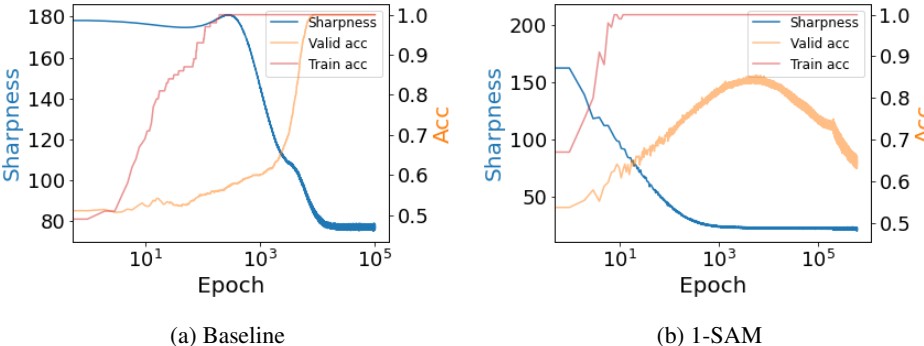

(a) Baseline          (b) 1-SAM

Figure 4: **Scenario II.** We train a 2-layer MLP with ReLU activation with Bias using gradient descent with weight decay and 1-SAM on $\mathcal{P}_{\text{xor}}$ with dimension $d = 30$ and training set size $n = 100$. One can clearly observe a distinction between the two settings. The minimum reached by 1-SAM is flatter but the model fails to generalize and the generalization performance even starts to degenerate after 4000 epochs. The difference between Figures 1b and 4b indicates a small change in the architecture can lead to a large change in the generalization performance.

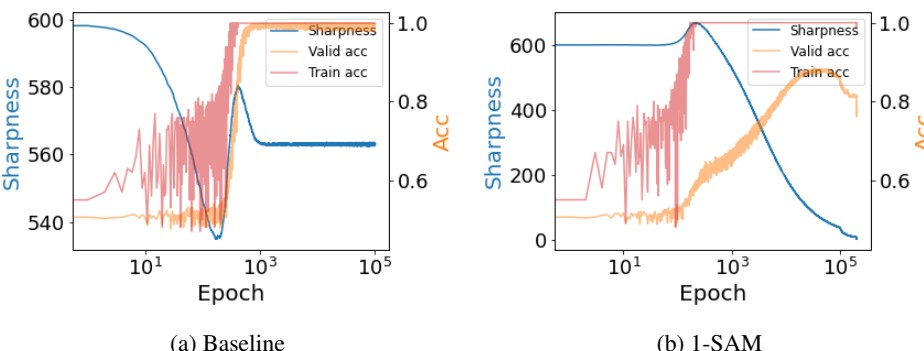

(a) Baseline          (b) 1-SAM

Figure 5: **Scenario II with Softplus Activation.** We train a 2-layer MLP with Softplus activation ($\mathsf{SoftPlus}(x) = \log(1 + e^x)$) with bias using gradient descent with weight decay and 1-SAM on $\mathcal{P}_{\text{xor}}$ with dimension $d = 30$ and training set size $n = 100$. We observe a similar phenomenon as Figure 4.

with $r = (d^2 + 1)^{1/4}$. then for any $x \in \{-1, 1\}^d$, it holds that $\mathsf{relu}(W_1 x + b_1) = r e_{5/2 - x[1] - x[2]/2}$ and $f_\theta(x) = x[1] \times x[2]$. Hence it is possible for Equation (5) to hold while the model has perfect generalization performance.

### 4.2 SAM empirically finds the non-generalizing solutions

In this section, we will show that in multiple settings, SAM can find solutions that have low sharpness but fail to generalize compared to the baseline full batch gradient descent method with weight decay. This proves that flat minimization can hurt generalization performance. However, one should note that Question 2 is not denied for the current architectures.

**Converged models found by SAM fail to generalize.** We perform experiments on data distribution $\mathcal{P}_{\text{xor}}$ in Figure 4. We apply small learning rate gradient descent with weight decay as our baseline and observe that the converged model found by SAM has a much lower sharpness than the baseline. However, the generalization performance of SAM is much worse than the baseline. Moreover, the generalization performance even starts to degenerate after 4000 epochs. We conclude that in this scenario, sharpness minimization can empirically hurt generalization performance.

**1-SAM may fail to generalize with other activation functions.** A natural question is whether the phenomenon that 1-SAM fails to generalize is limited to ReLU activation. In Figure 5, we show empirically that 1-SAM fails to generalize for 2-layer networks with softplus activation trained on the same dataset, although there is no known guarantee for the existence of memorizing solutions.

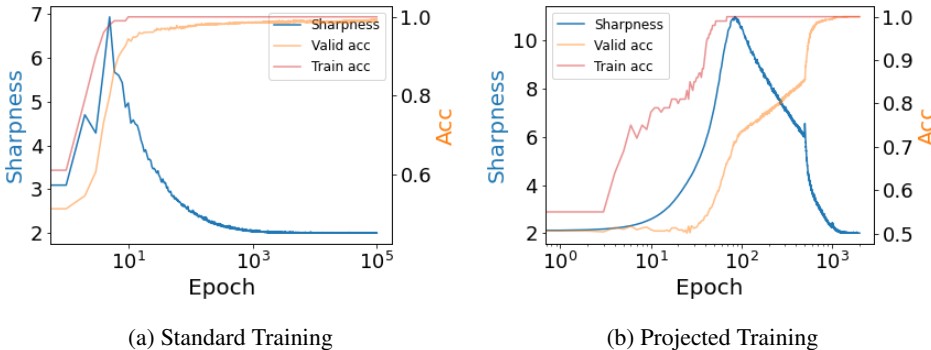

| (a) Standard Training | (b) Projected Training |

Figure 6: **Scenario III.** We train two-layer ReLU networks with simplified LayerNorm on data distribution $\mathcal{P}_{\mathrm{xor}}$ with dimension $d = 30$ and sample complexity $n = 100$ using 1-SAM. In Figure 6a, we use standard training. In Figure 6b, we restricted the norm of the weight and the bias of the first layer as 10, to avoid minimizing the sharpness by simply increasing the norm. We can see that in both cases, the models almost perfectly generalize.

## 5 Scenario III: Both Flattest Generalizing and Non-generalizing Models Exist, and SAM Finds the Latter

### 5.1 Both generalizing and non-generalizing solutions can be flattest

Despite the surprising contrary between Theorems 3.1 and 4.1, experiments show that Question 2 consistently hold. However, we will provide a counterexample in this section. Specifically, we will consider data distribution $\mathcal{P}_{\mathrm{xor}}$ and 2-layer ReLU MLP with simplified LayerNorm. One can first show both generalizing and non-generalizing solutions exist similar to Theorem 4.1 and propositions 4.1 and 4.2.

**Theorem 5.1.** *If the input data points $\{x_i\}$ of the training set form a set of extreme points (Definition 4.1), for sufficiently small $\epsilon$, then there exists a width-$n$ 2-MLP-Sim-LN with hyperparameter $\epsilon$ that is a memorizing solution (Definition 4.2) for the training dataset and has minimal sharpness over all the interpolating solutions.*

**Proposition 5.1.** *For data distribution $\mathcal{P}_{\mathrm{xor}}$, for sufficiently small $\epsilon$, for any number of samples $n$, there exists a width-$n$ 2-MLP-Sim-LN with hyperparameter $\epsilon$ that memorizes the training set as in Theorem 4.1, reaches minimal sharpness over all the interpolating models and has generalization error $\max\{1 - n/2^d, 0\}$ measured by zero one error.*

**Proposition 5.2.** *For data distribution $\mathcal{P}_{\mathrm{xor}}$, for sufficiently small $\epsilon$, for any number of samples $n$, there exists a width-$n$ 2-MLP-Sim-LN with hyperparameter $\epsilon$ that interpolates the training dataset, reaches minimal sharpness over all the interpolating models, and has zero generalization error measured by zero one error.*

The construction and intuition behind Theorem 5.1 and propositions 5.1 and 5.2 are similar to that of Theorem 4.1 and propositions 4.1 and 4.2. The proof is deferred to Appendix B.

### 5.2 SAM empirically finds generalizing models

Notice in Section 5.1 our theory makes the same prediction as in Section 4. However, strikingly, the experimental observation is reversed (Figure 6). Now running SAM can greatly improve the generalization performance till the model perfectly generalizes. This directly denies Question 2 as now we have a scenario in which sharpness minimization algorithms can improve generalization till perfect generalization while there exists a flattest minimizer that will generalize poorly.

## 6 Discussion and Conclusion

We present theoretical and empirical evidence for (1) whether sharpness minimization implies generalization subtly depends on the choice of architectures and data distributions, and (2) sharpness minimization algorithms including SAM may still improve generalization even when there exist flattest models that generalize poorly. Our results suggest that low sharpness may not be the only cause of the generalization benefit of sharpness minimization algorithms.

## ACKNOWLEDGEMENTS

The authors would like to thank the support from NSF IIS 2045685.

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
