# OpenReview forum: "Sharpness Minimization Algorithms Do Not Only Minimize Sharpness To Achieve Better Generalization"
_NeurIPS.cc/2023/Conference — NeurIPS 2023 oral_

### Official Review · Reviewer_ZroU · 2023-06-15

**Soundness:** 4 excellent
**Presentation:** 4 excellent
**Contribution:** 4 excellent
**Rating:** 8
**Confidence:** 4

**Summary:**

This paper sheds light on the relationship between three concepts: (a) generalization (b) flatness and (c) explicitly inducing flatness via SAM. The paper identifies that the relationship is nuanced and is based on certain architectural properties of the model.

 Specifically, depending on the architecture, there are three possible regimes:
1. A regime where _every flat minimizer generalizes well_ and SAM finds these minimizers.
2. A regime where _there exists flat minimizers that generalize poorly_ but SAM does _not_ find these minimizers.
3. A regime where _there exists flat minimizers that generalize poorly_ and SAM _does_ find these minimizers.

The paper shows that

- Regime (1) happens for 2-layer RELU networks _without_ bias
- Regime (2) happens for 2-layer RELU networks _with_ bias
- Regime (3) happens for 2-layer RELU networks _with_ bias, _with_ a simplified Layer Norm.

Note that in all cases, the existence of a flat solution that generalizes/memorizes is proven theoretically; while the effect of SAM is demonstrated empirically.

**Strengths:**

1. The questions formalized in the paper are novel, and well-formulated.
2. I gained a much clearer understanding of some generally unclear and confusing concepts: (a) flatness, (b) generalization and (c) explicitly inducing flatness. All these ideas are dealt with care and rigor.
3. The introduction is well-written and the chain of arguments are explained clearly. Despite being a theoretical and nuanced paper, I found it extremely easy to follow the arguments in each section. I thoroughly enjoyed reading the paper!
4. Rather than unnecessarily complicate the theory, the authors come up with a reasonably minimal settings of a 2-layer _non-linear_ model and support their arguments with either a formal result or with a rightly-designed empirical result.
5. That there are situations where the flattest solution does not generalize well is surprising from a theoretical point of view.

**Weaknesses:**

I currently do not have any major concerns about this paper. Most of my comments are follow-up questions, which I enumerate in the next section. My most important question --- which may or may not hint at a weakness --- is below:

### Major question

Q1a: Could you be more explicit about any parameter count dependence in the Rademacher complexity term of Thm 3.1 and Lemma 3.1? I am unsure if this is orthogonal to the main point of the result, so I'd also appreciate some clarity on that.

Q1b: On that note, I'd also like to know why this bound would break down upon the introduction of bias into the architecture. Would one of the terms within the Rademacher complexity end up scaling with the number of datapoints?

**Questions:**

1. Def 4.1 for the memorizing solutions appears to be incomplete and can be more formal. First, by _input data_, please clarify that you are referring to the training data. Furthermore, you also require the additional criterion that the activations of the model on _test_ data are all zero --- is this right? If not, the definition appears to be incomplete as even the raw inputs must more or less satisfy the orthogonality assumption with high probability? i.e., when we sample from a high-dimensional hypercube, two samples must more or less be orthogonal. So this alone shouldn't characterize a memorization solution?

2. Could you provide intuition for what the "flattest solution" in Eq 4 satisfies? It seems to require that _the activated top-layer weights have correspondence to the activations themselves_. I don't think Fig 3 captures this specific structure?

3. Do you empirically find that SAM learns a solution whose parameters resemble Eq 4 in any way?

4. The motivation behind the choice of loss function can be made more transparent. In particular, the main loss function used is squared error loss. In the appendix, the authors also use logistic loss _but with label smoothing_. If I am not wrong these choices ensure that the flattest minimizer exists and is unique up to some permutations. However, with pure logistic loss, the flattest minimizer would become ill-defined since there would be many different flat minimizers "at $\infty$"? I believe this is the reason behind a line of work in PAC-Bayes literature that tries to "normalize" flatness in a well-defined way. In other words, this work "sidesteps" this annoying issue; this is okay, as it still helps outline an elegant insight. But this needs to be clearly stated so that readers can put this result in context.

### Minor suggestions

1. I'd suggest making the specific notion of trace-of-hessian-based flatness clear in the abstract and in the beginning of the introduction. Since there are many notions of flatness, it would be good to be upfront about which notion is being discussed here.

2. In the introduction, I'd suggest clarifying that the SAM-related results are purely empirical.

3.  Section 4 title seems incorrect. In this regime, SAM finds the _latter_ non-generalizing solution not the former one.

4. Line 167 makes an abrupt reference to a theorem that's not discussed until then.

5. It's not clear until we get to the discussion of SAM as to whether the proof is empirical or not.

6. In Sec 5.1, instead of simply referring to Thm 3.1 and 4.1, I would add a verbal description so that it's easier to follow.


### Summary

Overall, I found this paper to be insightful, rigorous and clear in its message. The community would find this valuable given the confusing nature of literature in flatness.

### Post-rebuttal update

I have increased my score since the authors addressed my response satisfactorily.
**I'd also like to respectfully express my strong disagreement with concerns raised in the other reviews.**
- Regarding "Two-layer network is too simplistic": When a certain truth has not been proven in any setting to begin with, and when someone makes headway by providing the first proof in a simple setting, it is unfair and counter-productive to hold it against them. For without that headway, there is little hope for more general analyses. In fact, IMO proofs in more general settings (like a multi-layered network) get so unwieldy as to become uninsightful. Furthermore, in this scenario, the setting is a 2-layer network which does not reduce to a linear function of its parameters. IMO, the statements in this paper are already highly non-trivial to identify, let alone prove.
- Regarding considering other alternate versions of SAM, this is an interesting reformulation of the problem. Thanks to this work, we can be better posed to answer more important questions. Hence, again I don't want to fault this paper for not proving its elegant theory in various sophisticated versions of SAM. I would share this review's concern had the considered setting been far removed from practice, and promised no tangible way of extending to other situations --- this doesn't seem to be the case here, in my humble opinion.

**Limitations:**

The paper lists reasonable limitations in the conclusion.

---

> ### Author Rebuttal · Authors · 2023-08-10
>
> We appreciate the detailed review and comment! We would like to answer the reviewers’ questions here.
>
> 1. **Q1a.** Our bound is independent of the width of the model. Under the setting of Thm 3.1 and Lemma 3.1, we show our result by using sharpness as an upper bound for the parameter norm square using a uniform bound approach (see Lemmas A.5, A.6, and A.10).
> 2. **Q1b.** When the bias is introduced, both Lemma A.5 and A.6 will break. In this case, the sharpness can no longer be used as a valid upper bound for the weight norm and hence the flattest model will no longer fall into the function class defined in Lemma 3.1 with high probability.
> 3. **Questions 1.**  You are correct that ‘input data’ should be replaced by ‘training data’. We don’t need the activations on testing data to be all zero because our definition of memorizing solution only involves training data and does not necessarily imply bad generalization (c.f. Proposition 4.1, it takes some extra effort to show there exists a memorizing solution with bad generalization). Also, we would like to note that the construction of memorizing solution doesn’t need orthogonality within the input data. See Figure 3 as an example. We will try to clarify the statement in future versions. Thank you!
> 3. **Questions 2.** Here equation 4 characterizes the following correspondence: for each train data, among the activated neurons, the weights of the second layer are proportional to the activations (up to a factor of train data norm). The intuition behind this condition is that the 2-homogeneity of the network cause the gradient of the first and second layer weight ($W_1’$ and $W_2$) must have a constant product in the norm given an input and output pair. Equation 4 is then derived by balancing these two gradients. This proof technique could be useful for future theoretical analysis of homogeneous models.
> \
> Figure 3 and the memorizing solution we constructed is a special case where both sizes of equation 4 are vectors with only 1 nonzero coordinate. This can be done by choosing the first layer in a similar fashion as in Figure 3 which ensures all the input training data will activate exactly 1 neuron. Other than showing the activation is one-hot for each sample, figure 3 doesn't directly correspond to this additional structure, and shows no information about the second layer because the activated region of each hidden neuron is solely decided by its incoming weight, which belongs to the first layer.
>
> 4. **Questions 3.** Yes, the converged model in Figure 4b reaches almost minimal sharpness and we observe strong alignment in $\mathrm{relu}(W_1x_1 + b_1)$ and $W_2 \odot 1(W_1x_1 + b_1 > 0)$. The average cosine similarity between the two vectors is above 0.99.
>
> 5. **Questions 4.** We agree. We sidestep the infinity issue by using the soft label in the logistic loss. We will make clear the discussion here in future versions. Thank you!
>
> We would like to thank the reviewer again for the advice on writing and clarifying. We will adopt these suggestions in future versions.

---

> > ### Comment · Reviewer_ZroU · 2023-08-12
> > **Thanks for the response!**
> >
> > Thanks for the clear answers! I have raised my score and updated my review.

---

### Official Review · Reviewer_ex27 · 2023-06-27

**Soundness:** 2 fair
**Presentation:** 3 good
**Contribution:** 2 fair
**Rating:** 5
**Confidence:** 4

**Summary:**

This paper studies the connections between sharpness of the training loss and generalization performance under a simplified MLP architecture. In particular, authors show scenarios of when flat solution do/do not generalize, as well as cases when Sharpness-Aware Minimization (SAM) does/does not generalize.

**Strengths:**

The theoretical insights of the paper are interesting and hopefully can help better understand the relationship between flatness and generalization.

**Weaknesses:**

I think the paper has several shortcomings, outlined below.

- Model setup: I think the model setup is too simplistic. The input space is binary, which I think is fine. However, no noise is considered in the model and the output label $y=x[1]x[2]$ satisfies a very specific structure. Therefore, it is not clear to me why results discussed in the paper would generalize to broader model setups, and the observations discussed here are not just an artifact of the simplistic model. As shown in Lemma 3.1, the sharpness under the paper's setup is closely related to the weight norm, which may or may not be the case in a more realistic setup.

- Experiments: I don't find the experiments on SAM convincing. How is SAM implemented? How are the algorithm hyper-parameters chosen? How is SAM initialized? Maybe changing the SAM parameters and initialization would change the outcome of the experiments.  Can the observation of SAM converging to non-generalizing flat solution be replicated in a more realistic setup? Is weight decay used for SAM, as specially, in this case sharpness seems to be closely related to the weight norm.

- The authors miss [1]. As this paper discusses the connections between generalization and sharpness/SAM, the authors should discuss and compare their results to [1], specially as [1] discusses noisy setups.


[1] Behdin, K., & Mazumder, R. (2023). Sharpness-aware minimization: An implicit regularization perspective. arXiv preprint arXiv:2302.11836.

**Questions:**

- Maybe some visualization might help address some of my concerns in the previous part? For example visualizing the SAM/GD trajectories?

- Additional experiments either on noisy data settings or more realistic deep networks might help fill some of the gaps in the paper.

---

> ### Author Rebuttal · Authors · 2023-08-10
>
> We appreciate the review and comment! Regarding the reviewer’s concern, there seem to be some misunderstandings that we wish to clarify.
> 1. **W1: The generality of the results**
>
> While our positive result theorem 3.1 only holds for a 2-layer ReLU network without bias, our main results, which are negative for generalization for flattest models, hold for more general settings (see Theorem 4.1) including the noisy labels setting and multi-layer networks.
>
> 2. **W1: Relationship between sharpness and weight norm**
>
> We would also like to clarify that in all the setups excluding the 2-layer network without bias, sharpness is not necessarily related to weight norm. Lemma 4.1 only holds for data distribution with zero mean and a two-layer network without bias. For a two-layer network with bias, on the synthetic setup we consider, constraining the norm of the weight can actually induce generalization. Proposition 4.1 shows that constraining the sharpness can’t induce generalization, which shows that in this setup, sharpness is actually very different from the weight norm.
>
> 3. **W1: Noisy setups and additional related works**
>
> Experimentally SAM does have a generalization benefit over SGD when labels are noisy. But analyzing the generalization benefit of SAM for noisy labels is more complicated because the best performance is usually achieved by early-stopped SAM and interpolating the noise in the labels usually degrades the generalization. We leave understanding the generalization benefit of SAM with noisy labels as future work.
>
> The authors would like to thank the reviewer for pointing out the missing citation [1] and will add it in the following version. Compared with [1], our works are different in three ways:
>
> (a). [1] considers a two-layer relu net which is essentially linear in its trainable parameters, because the second layer is fixed to be all 1's, and the authors of [1] assume the activations of the first layer don't change throughout the training, while we consider a general two-layer relu net;
>
> (b). Because the model is linear as mentioned in (a) and the loss is l2 loss, all the solutions have the same sharpness (or more explicitly the same hessian) in [1], regardless of whether having noise. In contrast, in our setup, models can have different sharpness;
>
> (c). In the setting of [1], SAM and GD converge to the same solution and there is no generalization bound provided, regardless of having noise or not, while we show the flattest model in certain settings can always have good generalization (thm 3.1).
>
> 4. **Experiments.**
>
> a. **Weight decay.** We would like to stress that weight decay is not used for SAM because it is already proved that constraining the norm can lead to generalization in many setups (see Figure 4a for example) and our goal is to investigate the relationship between sharpness and generalization so adding weight decay may confound the relationship.
>
> b. **Hyperparameter.**  All the hyperparameters are listed in detail in Table 2 on page 30 of the appendix and we use standard initialization offered by Pytorch. SAM is implemented using an open-source library and the implementation can also be found in the appended material `code/sam.py`.
>
> Regarding the reviewer’s question, the authors perform a hyperparameter search on SAM for the experiments in Figure 4b. We search through learning rate from [3e-2, 1e-2, 3e-3, 1e-3] and perturbation radius from [0.01, 0.03, 0.05, 0.10]. We find out that using small initialization (scaling down the first layer by a factor of 0.01), the model can find a perfect generalizing solution using 1-SAM. We believe this is however caused by the implicit bias of small initialization, which might have a similar effect as norm constraint, as vanilla SGD also reaches good generalization performance using this initialization. However, using the standard initialization, the best model we found only reaches 0.84 validation accuracy and the hyperparameter is exactly the one we adopted in the paper. We believe this shows that what we have shown is not an artifact due to poor hyperparameter and we would also want to stress that we are only claiming that SAM can find a flattest but non-generalizing solution, which is inherently an existential claim.
>
> c. **Extension.** The authors have already done some extended experiments on other synthetic setups (see Figure 7 in the appendix) and also added an experiment on MNIST to further strengthen the paper according to the reviewer’s advice. Please see the general response for a detailed discussion.
>
> d. **Visualization.** We agree that visualization could help the readers better understand what happens during training. Unfortunately, we do not have a good idea of how to visualize a high dimensional trajectory except plotting certain statistics like train/valid acc/loss, sharpness, etc.

---

> > ### Comment · Reviewer_ex27 · 2023-08-14
> >
> > Thank you for your clarifications.
> >
> > - **The setup:** I still think the model setup is simple, but I also understand a proper analysis of noisy settings can be hard in general.
> >
> > - **Visualization:** Maybe https://arxiv.org/abs/1712.09913 can be helpful?
> >
> > - **Experiments:** I find the explanation provided satisfactory. Thank you!
> >
> > - **Comparison with [1]:** Thank you for your complete comparison.
> >
> > At the end, I think the work has some fairly important limitations in terms of model setup, but I acknowledge a general analysis can be hard. I will increase my score to 5.

---

### Official Review · Reviewer_2a5p · 2023-06-29

**Soundness:** 2 fair
**Presentation:** 2 fair
**Contribution:** 2 fair
**Rating:** 5
**Confidence:** 2

**Summary:**

This work aims to examine different cases for which flatness implies generalization. The conclusion heavily depends on the underlying model, while there are three different scenarios that contradict with each other. Additionally, the theoretical guarantees study the generalization error at the solution, however the empirical guarantees consider the generalization error at the last iteration of the SAM algorithm. Despite the convergence of the SAM algorithm, the training horizon may also affect generalization. With alternative sampling schedules of the data-set, it can be possible to minimize the training horizon (for instance see Repeated Random Sampling for Minimizing the Time-to-Accuracy of Learning by Okanovic et al.). I think it would be useful to examine if the SAM algorithm generalizes better when considering alternative sampling schedules. In such a case, the batch-selection policy could be more significant than the sharpness minimization. As the author also mention flattest solutions are not directly associated with efficient generalization in general.

Further, there is a deep literature for the edge of stability phenomenon (Gradient Descent on Neural Networks Typically Occurs at the Edge of Stability by Cohen et al.). In this case, the sharpness increases at termination (Understanding Gradient Descent on the Edge of Stability in Deep Learning by Arora et al., Section 2. Related Works), however good generalization performance have been observed in some cases. These cases might be considerd by the authors to provide a more complete picture for the empirical part of the paper, and additional conclusion about sharpness minimization and generalization.



**Strengths:**

The results are original and provide a partial evidence between sharpness and generalization, however the conclusion is not clear. Theorem 3.1 is an interesting result, however it is limited to a simple architecture but it is valuable.

**Weaknesses:**

Some parts of the paper are not clear enough. For instance, there exist statements like "All flattest models generalize" while the statement holds only for certain cases. It appears that the theoretical guarantees study generalization at the solution, however in practice the generalization of the algorithm might be different (see also the Summary). The main question of the paper has been considered in prior works as well.

**Questions:**

In theorem 3.1 the variable $\theta^*$ should be a random variable that depends on the training data-set. Is this true? In such a case the authors may consider to use the notation of the conditional expectation in the statement of the Theorem to make this fact clear.

**Limitations:**

The authors have addressed the limitations sufficiently.

---

> ### Author Rebuttal · Authors · 2023-08-10
>
> We thank the reviewer for the effort and time invested in the review. However, there are a lot of misunderstandings in the review and we clarify them below one by one.
> 1. To our best knowledge, there is no reported correlation between progressive sharpening and better generalization. Arora et al. 2022, shows that normalized GD reduces sharpness after reaching edge of stability and simultaneously improve generalization.
>
> 2. The goal of our work is to investigate sharpness and generalization. We agree with the reviewer that in practice many design choices including the batch sampling scheme can affect generalization but we choose to investigate the minimal setting where the data are i.i.d sampled and both the sharpness and the training loss are minimized. As shown in the paper, even this minimal setting would require much investigation.
>
> 3. We believe that our work has made a solid step toward revealing the relationship between sharpness and generalization by answering the questions listed in the introductions which are not answered by any previous works that the authors know of.
>
> 4. Regarding the title of the sections, the authors have already stated in the introduction that the relationship may subtly depend on architectures and data distributions. The authors have also clarified in both the introduction and section title that statements including `All flattest model generalize` can only be stated under the scenario defined by architectures and data distributions.
>
> 5. It is true that the variable is a random variable depending on the training data set. However, the randomness is already explicitly considered in `1 − δ over the random draw of training set`  on line 143.

---

> > ### Comment · Reviewer_2a5p · 2023-08-12
> > **Thank you!**
> >
> > Thank you for your response, I have raised the score because I think that examples of simple models are important, however I still believe that the presentation of the results and the paper requires improvement.

---

### Official Review · Reviewer_fkTA · 2023-07-06

**Soundness:** 3 good
**Presentation:** 2 fair
**Contribution:** 4 excellent
**Rating:** 7
**Confidence:** 3

**Summary:**

This paper explores the relationship between sharpness and generalization in overparameterized neural networks. The authors challenge the popular belief that flatness of the loss function implies better generalization and show that sharpness minimization algorithms do not always lead to better generalization. Through theoretical and empirical investigation, the authors identify three different scenarios for two-layer ReLU networks, showing that the relationship between sharpness and generalization is subtle and depends on the data distributions and the model architectures. The paper's contributions include a theoretical analysis of the relationship between sharpness and generalization.

**Strengths:**

The authors identify three scenarios for two-layer ReLU networks and provide theoretical and empirical evidence to support their findings. I believe the subtle details in architecture and distribution located in this paper are remarkable. Overall, this paper makes significant contributions to the field of generalization theory and challenges the popular belief that the flatness of the loss function implies better generalization.

**Weaknesses:**

1.	 One potential weakness of the paper is that it only examines the relationship between sharpness and generalization for two-layer ReLU networks (personally, I think this is acceptable since it is very hard to analyze more complex scenarios). Additionally, the authors only examine the CIFAR-10 and CIFAR-100 datasets, and it would be valuable to examine their findings on other datasets to determine if their results generalize to other domains.
2.	The expression of the article is not very friendly and a little difficult to understand.
3.	There are some grammatical and formatting problems, some of which are shown as follows.
1)	Pictures should be resized to make them look more coherent. In addition, the text in the image should be consistent with the font within the paper.
2)	There are some spelling errors, such as “subtlely”.
3)	References are poorly formatted. Please use a uniform format for references.
4)	Formulas and variables should be consistent with the font within the paper.
5)	Operators should be carefully defined.
6)	There are some formatting issues. For example, lowercase letters as the first letter of a sentence.


**Questions:**

1. In abstract, what does this sentence “there exist non-generalizing flattest models and sharpness minimization algorithms fail to generalize poorly” mean?

**Limitations:**

The authors have adequately addressed the limitations.

---

> ### Author Rebuttal · Authors · 2023-08-10
>
> We appreciate the reviewer’s positive review! Below are our responses to the reviewer’s concerns:
>
> * **W1.** We would like to clarify that we only perform experiments on synthetic tasks instead of CIFAR10 or CIFAR100 in the paper. However, we added an experiment on MNIST in the rebuttal phase. See the general response for a detailed discussion.
> * **W2.** Thank you for the helpful advice! We will fix and correct the formatting issues in future versions.
> * **Q1.** We made a typo in the quoted sentence — there should be no “poorly” at the end of the sentence. The corrected version is “there exist non-generalizing flattest models and sharpness minimization algorithms fail to generalize”.

---

### Author Rebuttal · Authors · 2023-08-10

We would like to thank the reviewers for the detailed reviews. Below we address a common question (by reviewers fkTA and ex27) on data distribution, whether the phenomena observed on synthetic datasets in this paper extend to vision datasets, by providing additional MNIST experiments.

**MNIST experiments**.
We observe that on the MNIST dataset when running SAM for a sufficiently long time, the validation accuracy will also drop while the sharpness decrease as in the synthetic setup, showing the generality of our result. We also observe that the converged accuracy of 1-SAM, in this case, is lower than the converged accuracy of SGD with weight decay regularization, which is also consistent with our findings in the synthetic setup.

Due to time constraints (we can only use batch size 1 for 1-SAM), we choose to downsample the MNIST image to 7$\times$7. We use a training sample size of 1000 and a 2-layer ReLU network with a width of 2000 here. The hyperparameters we choose and the resulting training curves are shown in the appended pdf.

Finally, we want to stress that even without additional vision experiments, our theoretical analysis has already excluded the possibility of proving a generalization bound for all flattest interpolating deep relu networks and shows that the relationship between flatness/sharpness minimization algorithm and generalization subtly depends on the architecture. For simplicity of presentation, we choose to focus on the two-layer net in the main paper and leave more studies on more realistic settings for future work.

---

### Decision · Program_Chairs · 2023-09-21

**Decision:**

Accept (oral)

**Comment:**

This paper provides a rigorous study of the relationship between flat minima, generalization, and the sharpness-aware optimizers such as SAM and presents intriguing and thought-provoking analyses/experiments/observations on the interaction of these three notions. Specifically, the authors show that whether flat minima are optimal or not (in terms of achieving the best generalization performance) depends on the choice of architecture and data distribution. In particular, they prove that for certain configuration of the above, the flattest minima provably correspond to the minima that generalize [and sharpness-aware optimizers can indeed find the flattest minima]. Perhaps more interestingly, the authors show that even when the flattest minima do not generalize well, SAM can still improve generalization. The latter observation is interpreted as there might be other regularization effects in SAM at work that are important for generalization and yet need to be understood.

All reviewers find the findings of this submission intriguing. and rate for accept. In concordance with them, I believe this paper is a timely research study into one of the biggest recent mysteries of generalization of deep neural networks, and hence recommend accept.